# Genome-Wide Identification and Expression Pattern Analysis of Dirigent Members in the Genus *Oryza*

**DOI:** 10.3390/ijms24087189

**Published:** 2023-04-13

**Authors:** Wen Duan, Baoping Xue, Yaqian He, Shenghao Liao, Xuemei Li, Xueying Li, Yun-Kuan Liang

**Affiliations:** 1State Key Laboratory of Hybrid Rice, Department of Plant Sciences, College of Life Sciences, Wuhan University, Wuhan 430072, China; duanwen@whu.edu.cn (W.D.); xuebaoping@whu.edu.cn (B.X.); 2018202040087@whu.edu.cn (Y.H.); 2020202040090@whu.edu.cn (S.L.); lixuemei@whu.edu.cn (X.L.); lixueying4221@whu.edu.cn (X.L.); 2Hubei Hongshan Laboratory, Wuhan 430070, China

**Keywords:** *Oryza*, dirigent gene family, phylogenetic analysis, expression profiles, environmental stress

## Abstract

Dirigent (DIR) members have been shown to play essential roles in plant growth, development and adaptation to environmental changes. However, to date, there has been no systematic analysis of the DIR members in the genus *Oryza*. Here, 420 genes were identified from nine rice species to have the conserved DIR domain. Importantly, the cultivated rice species *Oryza sativa* has more DIR family members than the wild rice species. DIR proteins in rice could be classified into six subfamilies based on phylogeny analysis. Gene duplication event analysis suggests that whole genome/segmental duplication and tandem duplication are the primary drivers for *DIR* genes’ evolution in *Oryza*, while tandem duplication is the main mechanism of gene family expansion in the DIR-b/d and DIR-c subfamilies. Analysis of the RNA sequencing data indicates that *OsjDIR* genes respond to a wide range of environmental factors, and most *OsjDIR* genes have a high expression level in roots. Qualitative reverse transcription PCR assays confirmed the responsiveness of *OsjDIR* genes to the undersupply of mineral elements, the excess of heavy metals and the infection of *Rhizoctonia solani*. Furthermore, there exist extensive interactions between DIR family members. Taken together, our results shed light on and provide a research foundation for the further exploration of *DIR* genes in rice.

## 1. Introduction

Dirigent (DIR) proteins, which were originally identified from *Forsythia intermedia*, guide the stereoselective coupling of coniferyl alcohol (CA) radicals in order to produce (+)-pinoresinol (a kind of lignan) [1]. DNA sequences homologous to the gene encoding the *F. suspensa* DIR protein have now been identified from bryophytes, ferns, monocotyledons, dicotyledons and other vascular plants including Arabidopsis [2], pepper [3], cotton [4], flax [5], mung bean [6] and *Isatis indigotica* [7]. Pinoresinol could be converted into other lignans including piperitol, laciresinol and secoisolaresinol [8]. Lignans are demonstrated to have crucial roles in plants’ pathogen defense responses [9,10]. Recently, the connections between DIR proteins and defense responses have been further substantiated. For example, the expression of a dirigent gene *GmDIR22* was up-regulated in the highly resistant soybean cultivar ‘Suinong 10′ when inoculated with *Phytophthora sojae*, and *GmDIR22* overexpression may improve resistance to *Phytophthora sojae* in soybeans [11]. The expression of *CaDIR7* was induced after the inoculation of *Phytophthora capsica*, and the silencing of *CaDIR7* damaged the defense against *Phytophthora capsica* in peppers [3]. The overexpression of GhDIR1 promoted lignan biosynthesis and blocked the spread of *Verticillium dahlia* in cotton [12]. The mutation of TaDIR-B1 increased lignan content and improved resistance to *Fusarium crown rot* in wheat [13].

Accumulating evidence suggests that the expression of many *DIR* genes is strongly affected by various abiotic stimuli. For example, a proteomic analysis showed that Mn toxicity induced DIR2-like protein levels and reduced the levels of another DIR protein in soybeans [14]. The transcription of a DIR gene from *Boea hygrometrica* (*BhDIR1*) was greatly enhanced under dehydration, cold conditions or heat stress conditions [15]. The expression of *TaDIR* in *Tamarix androssowii* was up-regulated after exposure to salt–alkali stress [16]. The mRNA level of *ScDir*, a DIR gene from sugarcane that is highly expressed in stems, has been reported to be increased by drought, high salt and oxidative stress [17]. Recently, the identification and genotypic analysis of the related mutants have revealed the indisputable roles of DIR proteins in lignification. In Arabidopsis, the dysfunction of a dirigent protein AtDIR10/ESB1 resulted in the disruption of CASP1 localization and Casparian strip (CS) formation, as well as ectopic deposition of lignification and suberin in the root [18]. The mutation of a dirigent protein ZmESBL in maize caused a compromised CS barrier, increased Na^+^ accumulation in the shoot and enhanced sensitivity to salt stress due to the hypo-lignification at endodermal CS [19]. Lignin deposition in the root cells of a specific type is important for the formation of root diffusion barriers that block the non-selective entry of solutes and water from the soil to the root stele and the leakage of mineral elements from the root back into the soil through the apoplastic transportation [20,21,22,23]. Nutrient availability from the rhizosphere also affects the lignification of root diffusion barriers [24,25]. An implication of these findings is that DIR proteins might participate in plant adaptation to the fluctuating concentrations of mineral elements in soil and to the heavy metal toxicity stress as well. However, experimental assays for the involvement of DIR proteins in these processes are currently lacking.

The genus *Oryza* contains twenty-two wild and two cultivated (*Oryza sativa* and *Oryza glaberrima*) rice species [26]. Zhu and coworkers found that genetic diversity in cultivated rice is only a quarter of that in wild rice due to the genetic drift caused by bottleneck effects [27]. Thus, some genes or alleles from wild rice that contribute to yield, stress tolerance and infection resistance still should be explored. Therefore, the genome-wide analysis of the DIR members in *Oryza* is important for obtaining beneficial DIR genes and thus for breeding improvements to crops. In this study, the phylogenetic relationship, chromosomal location, duplication events and selective forces of the DIR gene family in two cultivated rice species and seven wild rice species were first analyzed. Then, the expression patterns of DIR genes in response to the undersupply of mineral elements, the excess of heavy metals and the infection of *Rhizoctonia solani* were analyzed. In addition, the interaction of many DIR proteins was demonstrated by the yeast two-hybrid system. Taken together, these data provide a solid foundation for the further exploration of DIR genes in rice.

## 2. Results

### 2.1. Identification and Phylogeny Analysis of DIR Proteins

Based on the HMMER search (E-value ≤ 1 × 10^−5^, similarity > 50% as the threshold) and BLASTP (E-value ≤ 1 × 10^−5^ and an identity of 50% as the threshold), 48, 55, 36, 39, 40, 41, 44, 41, 43 and 33 DIR proteins were identified from *Oryza sativa* ssp. *Indica* (*Osi*), *Oryza sativa* ssp. *Japonica* (*Osj*), *Oryza glaberrima* (*Ogla*), *Oryza nivara* (*On*), *Oryza rufipogon* (*Or*), *Oryza barthii* (*Obar*), *Oryza glumipatula* (*Oglu*), *Oryza meridionalis* (*Om*), *Oryza punctata* (*Op*) and *Oryza brachyantha* (*Obra*), respectively (Table 1). As the data show in Table 1, *Obra* possesses the smallest number while *Osj* has the most DIR proteins among the rice species that we examined. The total numbers of DIR proteins in wild rice species are significantly less than those in cultivated rice *Oryza sativa*. For evolutionary relationship examination among rice species, a neighbor-joining (NJ) phylogenetic tree was constructed based on the 420 DIR protein sequence. DIR proteins in nine rice species could be divided into six subfamilies as indicated by Figure 1 This result agrees well with the previous reports [2,3,28]. We found that among these subfamilies, DIR-c has the largest number of DIR genes (90), which include 12 *OsiDIRs*, 15 *OsjDIRs*, 8 *OnDIRs*, 8 *OrDIRs*, 8 *ObarDIRs*, 9 *OglabDIRs*, 8 *OglumDIRs*, 13 *OmDIRs*, 4 *OpDIRs* and 6 *ObraDIRs*. DIR-b/d has 138 DIR genes from all of the examined rice species (Table 1). In contrast, DIR-a, e and g contain almost the same number of DIR genes: 60, 54 and 77, respectively (Table 1). Our results show that the number of DIR genes in DIR-b/d is greater than that in the DIR-a, e, f and g subfamilies. Interestingly, the number of DIR genes in the DIR-a, e, f and g subfamilies is nearly the same in different rice species. In contrast, the gene numbers in the DIR-c (4–15) and DIR-b/d (3–11) subfamilies among rice species are highly variable (Table 1 and Figure 1). The results suggest that the DIR genes in the DIR-a, e, f and g subfamilies were highly conserved among different rice species, while DIR-c and DIR-b/d expanded considerably in the rice species. In combination, these results indicate distinct dynamics of DIR gene family evolution in different rice species.

### 2.2. Gene Location and Duplication Events of DIR Proteins

As shown in Figure 2, the DIR genes are distributed on 12 chromosomes (Chr), while in comparison to other chromosomes, there are more DIR genes, and most of which form gene clusters on Chr7, Chr10 and Chr11 (Figure 2). It is worth noting that the majority of DIR genes existed at the ends of the chromosomes (Figure 2). Tandem duplication (TD) and whole genome duplication (WGD)/segmental duplication (SD) play a considerable role in the generation of gene families [29]. Therefore, we sought to analyze the duplication events of DIR genes in *Oryza*. In this study, 42 WGD/SD genes and 160 TD genes were identified in 2 cultivated and 7 wild rice species (Appendix A). Overall, in *Osi*, *Osj*, *On*, *Or*, *Obar*, *Ogla*, *Oglu*, *Om*, *Op* and *Obra*, 18 (37.5%), 17 (30.9%), 16 (41%), 13 (32.5%), 15 (43.9%), 12 (33.3%), 11 (25%), 14 (34%), 26 (60%) and 15 (46.9%) DIR genes were identified to be TD genes, and 6 (12.5%), 8 (9%), 3 (8%), 3 (7.5%), 4 (9.8%), 6 (16.7%), 6 (13.6%), 2 (5%), 4 (9%) and 0 (0%) DIR genes were identified to be WGD genes, respectively (Appendix A). These data suggest that TD and WGD/SD contributed to the expansion of *DIR* genes in *Oryza*, and TD played a predominant role. To further determine the evolutionary selection of DIR genes in *Oryza*, the nonsynonymous (Ka)/synonymous (Ks) ratios were counted using DnaSP (www.ub.edu/dnasp, accessed on 11 October 2021). The results displayed that the counted Ka/Ks value of most DIR duplication gene pairs was less than 1, whereas the Ka/Ks of the other gene pairs was more than 1 (Figure 3 and Appendix A). Moreover, as prompted by a previous report that Ks can be used to evaluate the appearance time of gene duplication [30], we analyzed and found that the divergence time of the DIR gene pairs in the genus *Oryza* was a range of 3.65 to 85.6 MYA (Figure 3 and Appendix A). In particular, the divergence time in *Or* was similar to that of *Om*, ranging from 7.47 to 85.6 MYA; the divergence time in the *Oglu*, *On*, *Obra* and *Obar* ranged from 3.65 to 71.23 MYA (Figure 3 and Appendix A). The divergence time in the *Op* and *Osi* ranged from 6.38 to 61.68 (Figure 3 and Appendix A). These results suggest that most duplicate genes of the DIR family were under purifying selection, and TD and WGD duplication events contributed chiefly to the evolution of the DIR gene family in the genus *Oryza*.

### 2.3. Synteny Analysis of DIR Genes in the Genus Oryza

To further understand the homology of the DIR gene family in the genus *Oryza*, we analyzed the collinearity of DIR genes between *Osj* and other rice species (Figure 4). Our results showed that the 36 *OsiDIR*, 32 *OnDIR*, 37 *OrDIR*, 19 *ObarDIR*, 31 *OglaDIR*, 36 *OgluDIR*, 38 *OmDIR*, 38 *OpDIR* and 31 *ObraDIR* homologous gene pairs had a collinearity relationship with *Osj* (Figure 4 and Appendix A). In *Oryza*, Chr11 has the largest syntenic gene pairs, and Chr7, Chr3, Chr1, Chr12 and Chr10 have only one or two homologous gene pairs (Figure 4; Appendix A). Significantly, we found that *OsjDIR5*, *OsjDIR9*, *OsjDIR17*, *OsjDIR26* and *OsjDIR50* were associated with at least two homologous gene pairs of other rice species, especially between *OsjDIR9* and the homologous gene pairs of *Osi*, *On*, *Or*, *Ogla*, *Oglu*, *Om*, *Op* and *Obra* (Figure 4 and Appendix A). The results suggest that these particular genes might have contributed to the occurrence and development of the DIR gene family.

### 2.4. Bioinformatics Analysis of the Expression Patterns of OsjDIR Genes

To delve into the potential regulatory mechanisms of the *OsjDIR* genes, a 2k promoter region of the *OsjDIR* genes was scanned for the putative transcription factor binding sites using the TRANSFAC database (http://jaspar.binf.ku.dk/, accessed on 15 may 2022). A total of 781 DNA-binding motifs of transcription factors were identified (Appendix A). The promoters of the majority of the *OsjDIR* genes have transcription factors binding sites related to biotic stress- (30), abiotic stress- (16), hormone- (15) and development (19) (Appendix A). Furthermore, we also found that the transcription factor binding sites associated with secondary cell wall formation and light response existed in the promoter of the *OsjDIR* genes (Appendix A). The respective promoter of *OsjDIR21*, *OsjDIR29*, *OsjDIR35*, *OsjDIR36* and *OsjDIR37* contains binding motifs of transcription factors such as DREB1A, WRKY25, WRKY26, CEJ1, ERF2, WRKY18 and WRKY29 (Appendix A), which have been well documented to regulate gene expression related to drought stress, salt stress, heat stress and pathogen infection [31,32,33,34]. These results indicate that the transcription of these *DIR* genes might respond to drought, salt, heat stress and pathogen infection. Recently, transcription factors MYB46, ERF38, MYB52 and NST3 have been identified as the primary regulators of secondary cell wall formation [35,36]. We found that the promoter of *OsjDIR22* and *OsjDIR33* have the binding sites of MYB46 and ERF38 and the promoter of *OsjDIR49* has the binding sites of ERF38 (Appendix A). These results indicate that *OsjDIR* genes might participate in plant growth, development and adaption to ever-changing environmental conditions. Then, the Plant Public RNA-seq Database (PPRD, http://ipf.sustech.edu.cn/pub/plantrna/, accessed on 11 July 2022) was used to mine the available transcriptome microarray data regarding the expression of *OsjDIR* genes in root, flag leaf, stem and flower (Figure 5 and Appendix A) [37]. According to our analysis, 30 *OsjDIR* genes have a high expression in the root, whereas only very few *OsjDIR* genes have the highest expression level in other organs (e.g., *OsDIR44* in the flag leaf and *OsDIR5* in the stem) (Figure 5 and Appendix A). In addition, we further downloaded the RNA sequencing (RNA-seq) data from PPRD to analyze the expression profiles of rice DIR genes under heat, salt, drought, mineral deficiency stress and pathogen infection. Among 55 *OsjDIR* genes, 10 *OsjDIR* genes could not be detected in the RAN-seq data. As shown in Appendix A, when plants suffered from low N, Mn deficiency, Cu deficiency, Zn deficiency, Fe deficiency, osmotic stress, flood stress, cold stress, Xoo and sheath blight infection, 33, 33, 25, 27, 33, 15, 18, 14, 5 and 7 *OsjDIR* genes were down-regulated, and 4, 12, 18, 16, 8, 8, 9, 8, 11 and 14 *OsjDIR* genes were up-regulated, respectively (Appendix A). The expression of *OsjDIR4* was up-regulated after osmotic treatments (Appendix A). The transcription of *OsjDIR36* was greatly up-regulated (more than 10-fold) in the Jingang cultivar when inoculated with sheath blight; however, no such up-regulation was detected in the Yanhui cultivar (Appendix A). Both of the expressions of *OsjDIR30* and *OsjDIR31* were down-regulated (more than 100-fold) under iron deficiency compared with mock treatment (Appendix A). These results evidently indicate that DIR genes are involved in plant response to a wide range of environmental factors (including drought, cold and nutrient elements) and biotic agents of plant diseases.

### 2.5. Experimental Examinations of the Expression Dynamics of OsjDIR Genes

According to the PPRD RNA-seq data, most *OsjDIR* genes have an expression level in the root, implying that these genes might participate in nutrient uptake. Therefore, we conducted qRT-PCR tests to examine the expression dynamics of *OsjDIR* genes under nutrient element deficiency and heavy metal stress. As shown in Figure 6A, the transcriptions of *OsjDIR13*, *OsjDIR25* and *OsjDIR29* were induced by low-nitrogen treatments, while the mRNA levels of *OsjDIR11*, *OsjDIR15*, *OsjDIR21*, *OsjDIR24*, *OsjDIR30*, *OsjDIR41* and *OsjDIR46* were down-regulated under treatments with low nitrogen (Figure 6A). When phosphate supply was deficient, the transcript levels of *OsjDIR3*, *11*, *13*, *21*, *22*, *30*, *31* and *32* were all induced, whereas those of *OsjDIR15*, *24*, *29* and *33* were suppressed (Figure 6A). Intriguingly, several *OsDIR* genes showed the opposite expression patterns under low-nitrogen and low-phosphate treatments. For example, the expressions of *OsjDIR11*, *21* and *30* were repressed by low-nitrogen but induced by low-phosphate stress, whereas that of *OsjDIR29* was induced by low-nitrogen but repressed by low-phosphate treatments (Figure 6A). In addition, *OsjDIR11*, *OsjDIR35*, *OsjDIR41* and *OsjDIR53* showed significant up-regulation under calcium-deficiency stress; *OsjDIR11*, *15*, *22*, *24*, *29*, *33*, *35*, *41*, *49* and *53* displayed a marked up-regulation under copper-deficiency stress. In contrast, seven *OsjDIR* genes (*OsjDIR3*, *11*, *15*, *22*, *24*, *33* and *48*) all showed a significant down-regulation under high cadmium conditions (Figure 6B). The results indicate that *OsjDIR* genes participate in rice responses to various nutrient element deficiency and heavy metal stress.

### 2.6. Expression of OsjDIR Genes to Rhizoctonia solani Inoculation

To explore the possible role of *OsjDIR* genes in plant defense, we next determined the expression responses of 10 *OsjDIR* genes to *Rhizoctonia solani* (*R. solani*), a globally distributed phytopathogenic fungus with a diverse host range that causes sheath blight in rice, maize and other Gramineous plants [38,39]. Our results showed that the mRNA levels of most *OsjDIR* genes were down-regulated by the treatment. For example, the mRNA levels of *OsjDIR13*, *15*, *21*, *29*, *35*, *36* and *37* were significantly lower at 12, 24 and 48 h post treatment with reference to the control (Figure 7). Additionally, note that the transcript abundance of the *OsjDIR12* gene was significantly up-regulated at 12 h after the inoculation of *R. solani*, whereas it was significantly down-regulated at 24 and 48 h post inoculation (Figure 7).

### 2.7. Many DIR Proteins Interact with Themselves or Other DIR Proteins

Considering the tightly trimeric crystal structure of DIR proteins, which was reported in pea (*Pisum sativum*) and Arabidopsis [40,41], we also sought to predict the 3D structure of DIR proteins through the SWISS website (https://swissmodel.expasy.org/interactive, accessed on 5 April 2022). As a result, 55 OsjDIR proteins in *Osj* were all predicted to be able to form a trimer structure (Appendix A). Next, a protein–protein interaction network among DIR proteins in Arabidopsis was constructed using STRING (https://cn.string-db.org/, accessed on 20 April 2022). The results indicated extensive interactions between DIR proteins (Appendix A). To experimentally validate the predicted interactions between the DIR proteins in rice, we subjected OsjDIR 30, 32, 34, 35 and 38 to the yeast two-hybrid assays (Y2H). As the data show in Figure 8, we found that OsjDIR30, 32 and 38 can interact with themselves, and OsjDIR32 can interact with OsjDIR34 and OsjDIR35 (Figure 8). These observations imply that DIR proteins might perform their functions either independently or through interacting with others for proper biological activity in rice.

## 3. Discussion

DIR proteins are ubiquitously presented in vascular plants, including ferns, gymnosperms and angiosperms. For example, 26, 24, 19 and 107 DIR genes have been identified from Arabidopsis, pepper, cotton and *Isatis indigotica*, respectively [2,3,4,7]. However, *DIR* genes in *Oryza* have so far received little attention. In this study, we conducted a systematic bioinformatics analysis of the *DIR* gene family in *Oryza*, and identified 48, 55, 39, 40, 41, 36, 44, 41, 43 and 32 *DIR* genes from *Osi*, *Osj*, *On*, *Or*, *Obar*, *Ogla*, *Oglu*, *Om*, *Op* and *Obra*, respectively (Table 1). Interestingly, the number of *DIR* genes (32) in *Obra* was much less than that in *Osi*, *Osj*, *On*, *Or*, *Ogla*, *Oglu*, *Om* or *Op* (Table 1). We hypothesized that the smallest genome of *Obra* among the rice species might (at least partially) account for the difference [42]. In this study, the number of *DIR* genes in the cultivated rice species *Oryza sativa* was more than that in all of the wild types we tested (Tabel 1), fitting well with a recent report that TD and SD/WGD might be the main driving forces in the expansion of the *G. barbadense* DIR gene family [4]. As demonstrated in Table 1 and Figure 2, 6 SD/WGD genes and 18 TD genes were identified in *Osi*, and 8 SD/WGD genes and 17 TD genes were identified in *Osj*, whereas no SD/WGD was identified in *Obra* (Table 1 and Figure 2). The numbers of SD/WGD genes and TD genes in *Osi* and *Osj* were greater than those of wild rice, except for the *Op* (Table 1 and Figure 2). These results indicate that the DIR gene family has expanded in the long domestication process and the SD/WGD and TD events might account for the expansion and evolution of the DIR gene family in cultivated rice *Oryza sativa*. Forty-three DIR genes were identified in *Op,* far less than in *Osj* (fifty-five) or *Osi* (forty-eight), but there were twenty-six TD genes in *Op* (Table 1; Figure 3). These results suggest that distinct mechanisms underlying *DIR* gene expansion might exist in the genus *Oryza* and the *OsDIR* genes might have other types of expansion modes, such as proximal and replicative transposition, which merit further investigation.

The phylogenetic tree analysis suggested that 420 DIR proteins were divided into 6 subfamilies (Figure 2). Compared with the other subfamilies, both DIR-b/d (88) and DIR-c (90) subfamilies possessed many more *DIR* genes (Table 1 and Figure 2). Consistent with this observation, the tandem duplication genes are relatively over-dominant in the DIR-b/d and DIR-c subfamilies (Table 1 and Figure 3), suggesting that there is a certain trend in the expansion of genes in the DIR-b/d and DIR-c subfamilies. The similar results recently found in other plants, including pepper, cotton, spruce and flax, have partly borne out this suggestion [3,28,43]. Note also that tandem duplication in plants has been deemed to be an appropriate response to the adaption of the changing environment [44]. Therefore, the rapid expansion of the DIR-b/d and DIR-c subfamilies might be available for improvement in stress tolerance. In addition, our results showed that the Ka/Ks of most *DIR* gene pairs in *Oryza* was not more than 1 (Figure 3), suggesting that the duplication gene pairs were under purifying selection in *Oryza*, fitting well with the studies into cotton and other plants [4].

Sixty percent of *AtDIR* genes display the highest expression in roots, whereas only a few of them are preferentially expressed in other tissues [2]. With good agreement, our results show that about half of the *OsjDIR* genes are expressed with a maximum level in rice roots (Figure 5). In Arabidopsis, the mutation of a dirigent domain-containing protein AtDIR10/ESB1 caused increased shoot concentrations of Na, S, K, As, Se and Mo and decreased shoot concentrations of Fe, Ca, Mn and Zn compared with wild-type plants [45]. Subsequent studies have shown that AtDIR10/ESB1 is involved in regulating the Casparian strip development in root [18], suggesting that DIR proteins might affect ion absorption and mineral nutrient homeostasis. Consistently, in this study we demonstrated that many *OsjDIR* genes are sensitive to mineral deficiency and heavy metal stress (Figure 6). Notably, *OsjDIR11*, *OsjDIR21*, *OsjDIR30* and *OsjDIR29* showed opposing expression patterns under low-nitrogen and low-phosphate treatments (Figure 6A). On the contrary, *OsjDIR13* and *OsjDIR24* showed similar expression patterns under low-nitrogen and low-phosphate treatments (Figure 6A). Previous studies showed that the proper N:P supply ratio in wetland graminoids was essential for enhancing the uptake of P [46,47]. However, the mechanisms of N–P cross-talk remain unknown. Therefore, *OsjDIR11*, *OsjDIR21*, *OsjDIR30*, *OsjDIR29*, *OsjDIR13* and *OsjDIR24* might be the candidate genes that coordinate the utilization of N and P.

In wheat, a *DIR* gene has been confirmed to be involved in the pathogen defense. This work showed that *TaDIR-B1-silenced wheat lines* accumulated more lignin and showed higher resistance to *fusarium crown rot* than wild types, suggesting that the *TaDIR-B1* gene negatively regulated the *fusarium crown rot* resistance in wheat [13]. Sheath blight (ShB) is one of the highly destructive diseases of rice, and sheath blight infection usually leads to decreases in rice yield [38]. According to our qRT-PCR results, the expressions of *OsjDIR13*, *OsjDIR15*, *OsjDIR21*, *OsjDIR29*, *OsjDIR35*, *OsjDIR36* and *OsjDIR37* were significantly suppressed by *Rhizoctonia solani* inoculation (Figure 7). It is worth noting that *OsjDIR35*, *OsjDIR36* and *OsjDIR37* are tandem duplication genes, implying their possible synergistic roles in regulating rice resistance to sheath blight infection.

Our observations show that many DIR proteins can interact with each other in rice during the yeast two-hybrid assays (Figure 8), suggesting that the DIR proteins might perform their functions either independently or through interacting with others. This suggestion is compatible with the model put forwarded by Gasper and coworkers, which shows Arabidopsis AtDIR6 as being an eight-stranded antiparallel β-barrel that forms a trimer with the well-spaced vacancies cavity for substrate binding [41]. Our preliminary data of the predicted 3D structure of DIR proteins and the Y2H assays lend strong support to the trimer structure model in which two substrate radicals bind to each of the DIR monomers. More research is required for a more refined understanding of the function of the DIR genes in rice.

## 4. Materials and Methods

### 4.1. Identification of DIR Genes in the Rice Species

The genomes and the genomic data of *Osi*, *Osj*, *Ogla*, *Oruf*, *Op*, *On*, *Om*, *Oglu*, *Obar* and *Obra* were downloaded from Ensembl Plants (http://plants.ensembl.org/index.html/, accessed on 2 September 2021). Genome sequences of *Osi* were derived from a rice cultivar 9311; genome sequences of *Osj* were derived from a rice cultivar Nipponbare; genome sequences of *Ogla* were derived from *O. glaberrima* Steud; genome sequences of *On* were derived from *O. nivara* Sharma et Shastry; genome sequences of *Or* were derived from *Oryza rufipogon* W. Griffith; genome sequences of *Obar* were derived from *Oryza barthii* A. Chev.; genome sequences of *Oglu* were derived from *O. glumipatula* Steud; genome sequences of *Oryza meridionalis* were derived from *Oryza meridionalis* N.Q.Ng; genome sequences of *Oryza punctata* were derived from *Oryza punctata* Kontshy ex Steud and genome sequences of *Oryza brachyantha* were derived from *Oryza brachyantha* A. Chev et Rocher. The DIR protein sequence of *Arabidopsis thaliana* was downloaded from TAIR (https://www.arabidopsis.org/, accessed on 5 September 2021). The dirigent domain (PF03018) was downloaded from Pfam (http://pfam.sanger.ac.uk/, accessed on 8 September 2021). HMMER 3.0 was used to search the DIR proteins from different rice species’ genome assemblies with E-value ≤ 1 × 10^−5^ and similarity > 50% as the thresholds. Furthermore, the DIR protein sequences of Arabidopsis were used as the references to find the DIR proteins in the 9 rice species using a BLASTP-method-based search (E-value 1 × 10^−5^ and an identity of 50% as the thresholds). After removing the redundant sequence, all candidate DIR protein sequences were submitted to NCBI-CDD (https://www.ncbi.nlm.nih.gov/Structure/bwrpsb/bwrpsb.cgi, accessed on 10 September 2021) and SMART (http://smart.embl-heidelberg.de/, accessed on 10 September 2021) for confirming the conserved DIR domain. The obtaining of the longest transcripts was achieved using R package seqfinder (https://github.com/yueliu1115/seqfinder, accessed on 13 September 2021).

### 4.2. Phylogenetic Analysis of DIR Proteins in the Different Rice Species

The dirigent proteins identified in the nine rice species were used for phylogenetic analysis. The phylogeny tree was generated using the neighbor-joining (NJ) method of MEGA7.0, with 1000 bootstrap replications [48].

### 4.3. Gene Duplication Analysis and the Calculation of Ka/Ks Ratios

The MCScanX (with default parameters) was used to detect segmental and tandem duplications [49]. Subsequently, the nonsynonymous (Ka)/synonymous (Ks) ratios were analyzed using TBtools for selective force analysis [50]. Moreover, the divergence time was estimated by T = Ks/(2 × 9.1 × 10^−9^) × 10^−6^ million years ago (MYA) [30].

### 4.4. Expression Patterns of Rice DIR Genes through Analyzing the RNA-Seq Data

Expression patterns of *DIR* genes were downloaded from PPRD (http://ipf.sustech.edu.cn/pub/plantrna/, accessed on 11 July 2022) [51]. Log_2_ (1 + FPKM (treatment)/1 + FPKM (control)) were calculated using FPKM to indicate the fold change in the gene expression level. TBtools was used to generate the heatmap [50].

### 4.5. Plant Materials and Growth Conditions

Rice cultivar Zhonghua 11 (*Oryza sativa* ssp. *japonica*, ZH11) developed by Institute of Crop Sciences, Chinese Academy of Agricultural Sciences, was used in this study. Rice seedlings were cultivated in a greenhouse with the condition of 12/12 h light/dark (200 μmol m^−2^ s^−1^), 28 °C and 70% humidity. In the greenhouse, the newly germinated seedlings were planted on top of the custom-made black containers with the modified Kimura B solution (see Appendix A) [47] to allow the roots to grow in the dark. For the vigorous growth of seedlings, the nutrient solution was replaced four times a week.

### 4.6. Plant Materials’ Treatments

For low-P treatment, 10-day rice seedlings cultured with Kimura B solution were transferred into Pi sufficient (0.18 mM KH_2_PO_4_) or low P (0.018 mM KH_2_PO_4_) for 7 days. The roots were sampled at 7 days post treatment. For low-N treatment, 10-day rice seedlings cultured with Kimura B solution were transferred into HN (5 mM KNO_3_) or LN (0.2 mM KNO_3_) for 7 days. The roots were sampled at 2 and 7 days post treatment. For nutrient element deficiency and high CdCl_2_ treatments, rice seedlings were cultured with Kimura B solution for 7 days and then supplemented with 0 μM CaCl_2_, 0 μM CuSO_4_, 0 μM Fe(II)-EDTA and 50 μM CdCl_2_ for 2 weeks. The roots were sampled on Day 7 post treatment. For the infection of *Rhizoctonia solani*, the *R. solani* AGI-IA strain was cultured on potato dextrose agar (PDA) medium [45]. After two-day growth, the medium was cut off by a puncher and placed on the leaf sheath; then, the rice plants were grown under a 14/10 h light/dark photoperiod at 25 °C for 12 h, 24 h and 48 h.

### 4.7. RNA Extraction and qRT-PCR Analysis

Primers for the *OsjDIR* genes and *OsActin* in this study are shown in Appendix A. The extraction of Total RNA and the synthesis of cDNA were performed using a KKFast Plant RNApure Kit (ZOMANBIO) and PrimeScript™ RT reagent kit (TaKaRa), respectively. The qRT-PCR reaction system (10 μL) contained 5μL (2 × SYBR qPCR Mix, ZOMANBIO), 3μL ddH2O, 1 μL primer (F + R) and 1 μL cDNA. The qRT-PCR reaction program was set as follows: 95 °C for 60 s, followed by 40 cycles of 95 °C/10 s and 60 °C/30 s, using the Bio-Rad 384 wells system. Three biological repeats were set for each sample. The average cycle (Ct) was used for the relative quantitation via the 2^−∆∆CT^ method.

### 4.8. Yeast Two-Hybrid Assays

The coding sequences of the OsDIR30, 32, 34, 35 and 38 were cloned into the pGBKT7 vector (BD-DIRs) or PGADT7 vector (AD-DIRs). The positive (AD-T + BD-53) and negative controls (AD-T + BD-Lam) were offered by the manufacturer Clontech. Yeast transformants with empty PGADT7 and BD-DIRs and yeast transformants with AD-DIRs and empty pGBKT7 grown on tryptophan, leucine and histidine removed medium with 1 mM 3-AT were also used as the negative controls. Yeast transformants with AD-DIRs and BD-DIRs grown on tryptophan, leucine and histidine removed medium with 1 mM 3-AT were used to identify the interaction between the DIR members.

### 4.9. Statistical Analysis

All dates are the means ± SD of three biological replicates. Significant differences (*p* < 0.01) were determined via one-way analysis of variance and Tukey’s multiple comparisons test using Prism 8 software.

## 5. Conclusions

In this study, we first identified 420 DIR proteins from the 2 cultivated and 7 wild rice species and grouped them into 6 subfamilies. The number of DIR genes in the cultivated rice species *Oryza sativa* was more than that in all of the wild types that we tested. Then, we showed that TD and SD/WGD duplication events were the major driving forces for the expansion of *DIR* genes in the cultivated rice species *Oryza sativa*. In addition, the analysis of the RNA sequencing data indicated that *OsjDIR* genes responded to a range of stresses, and most *OsjDIR* genes had a high expression level in roots. qRT-PCR assays confirmed that *OsjDIR* genes responded to the undersupply of mineral elements (N, P, Fe, Cu and Ca), the excess of heavy metal (Cd) and the infection of *Rhizoctonia solani*. A yeast two-hybrid assay indicated that DIR proteins might perform their functions either independently or through interacting with others for proper biological activity in rice. In conclusion, our results shed light on and provide a research foundation for the further exploration of the DIR genes in rice. 

## Figures and Tables

**Figure 1 ijms-24-07189-f001:**
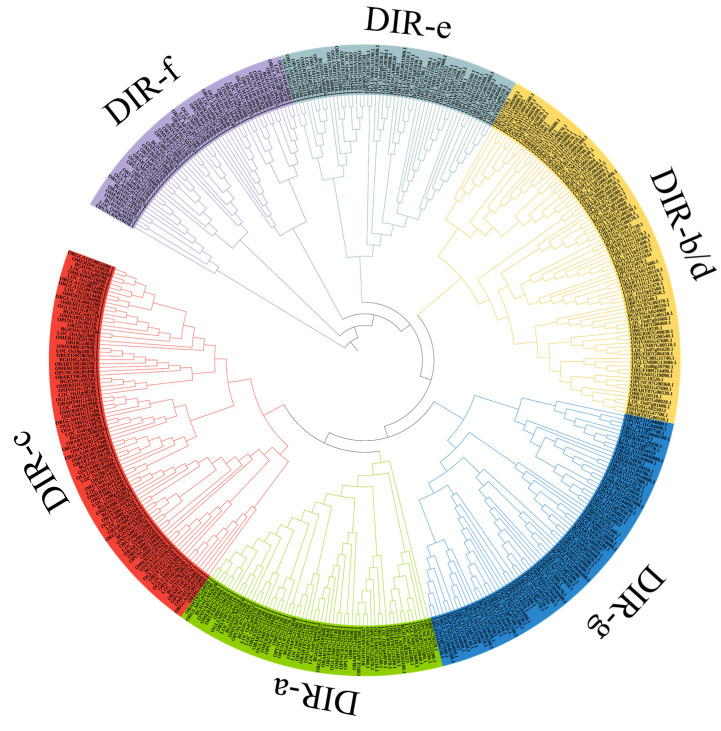
Phylogenetic analysis of dirgent (DIR) proteins from nine rice species. MEGA 7.0 was used to carry out protein sequence alignment and subsequently the bootstrap tree construction with the neighbor-joining (NJ) method and 1000 bootstrap value setting. The different DIR subfamilies are indicated by green (**a**), yellow (**b**/**d**), red (**c**), light blue (**e**), purple (**f**) and blue (**g**), respectively. For additional information, see Appendix A.

**Figure 2 ijms-24-07189-f002:**
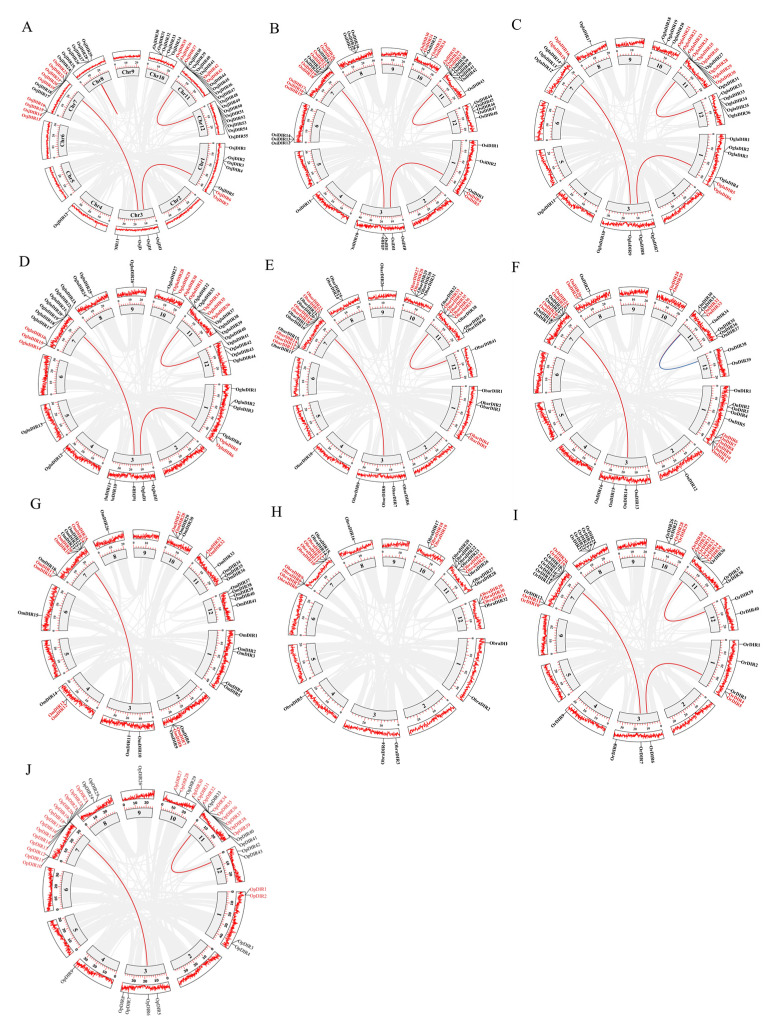
The location and duplication events of the DIR genes in the genus *Oryza*, including (**A**) *Osj*; (**B**) *Osi*; (**C**) *Ogla*; (**D**) *Oglu*; (**E**) *Obar*; (**F**) *On*; (**G**) *Om*; (**H**) *Obra*; (**I**) *Or* and (**J**) *Op*. Genes with red color represent the tandem duplicated genes and genes with red line represent the whole genome/segmental duplication genes.

**Figure 3 ijms-24-07189-f003:**
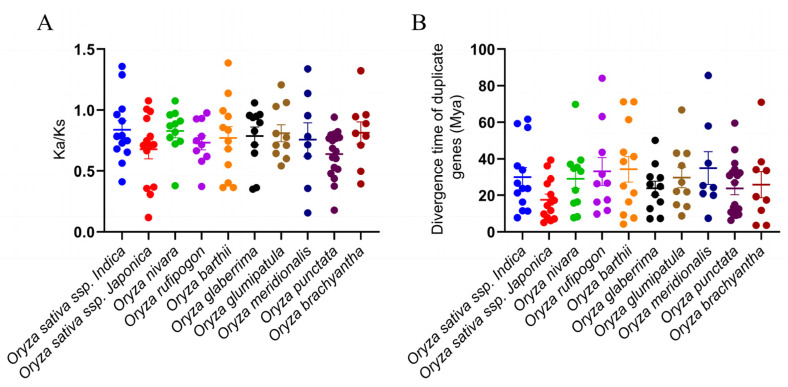
Selection pressure and divergence time analysis of DIR gene pairs in the nine rice species. (**A**) Ka/Ks ratio calculation of the DIR gene pairs in the nine rice species. (**B**) The divergence time prediction of DIR gene pairs in the nine rice species.

**Figure 4 ijms-24-07189-f004:**
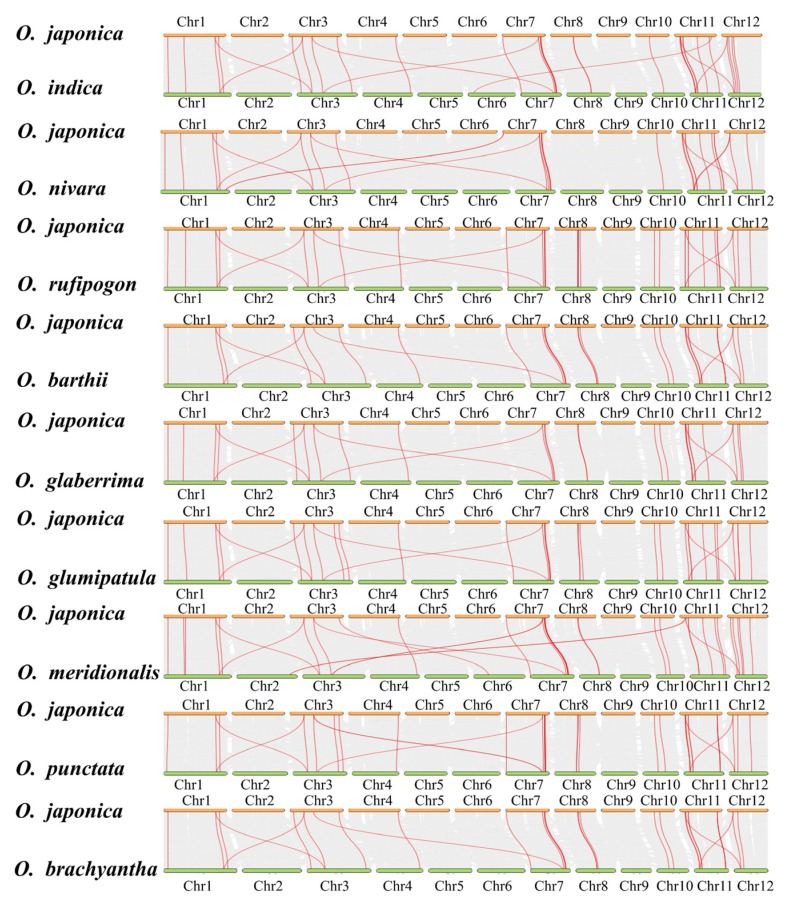
Synteny analysis of DIR genes between *Osj* and other rice species. The collinear blocks between *Osj* and other rice species are shown by gray lines. The syntenic DIR gene pairs between *Osj* and other rice species are highlighted by red lines.

**Figure 5 ijms-24-07189-f005:**
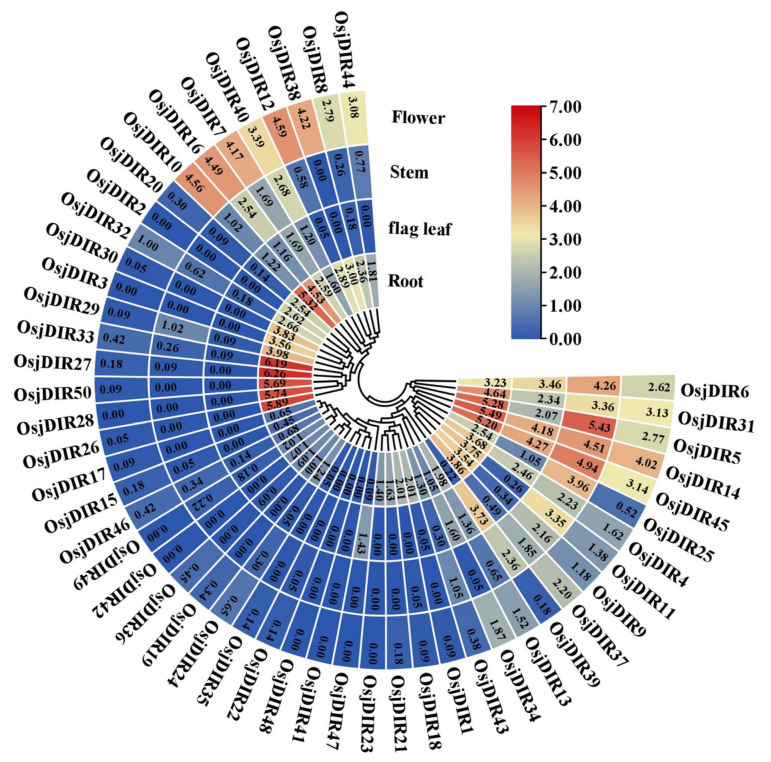
The heatmap of the expression pattern of *OsjDIR* genes in the root, flag leaf, stem and flower based on RNA-seq data. Log_2_ (1 + FPKM (treatment)/1 + FPKM (control)) were calculated using FPKM to indicate the fold change in the gene expression level. TBtools was used to generate the heatmap.

**Figure 6 ijms-24-07189-f006:**
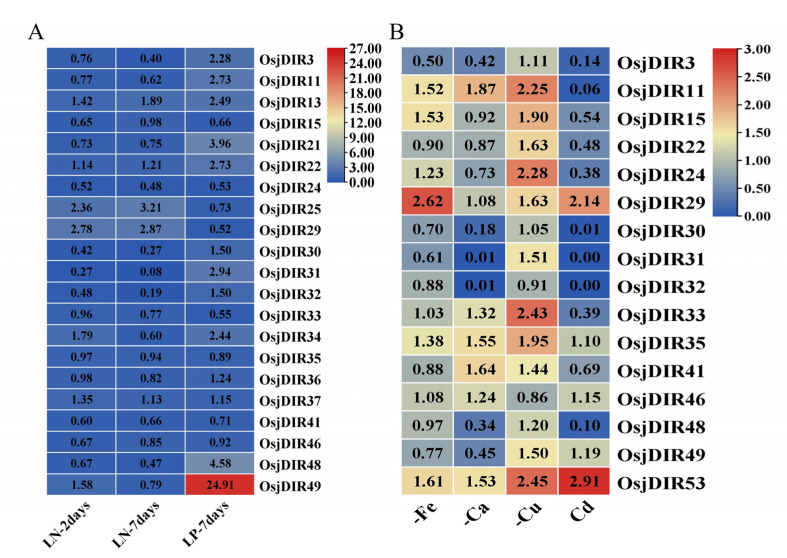
Expression patterns of *OsjDIR* genes in response to nutrient element deficiency and high CdCl2 stress. (**A**) Expression profiles of *OsjDIR* genes in response to low-nitrogen (LN) and low-phosphate (LP) treatments. (**B**) Expression profiles of *OsjDIR* genes’ nutrient element deficiency (-Fe, -Ca and -Cu) and high CdCl2 stress treatments. qRT-PCR analysis was performed using the rice *OsActin* as an internal control.

**Figure 7 ijms-24-07189-f007:**
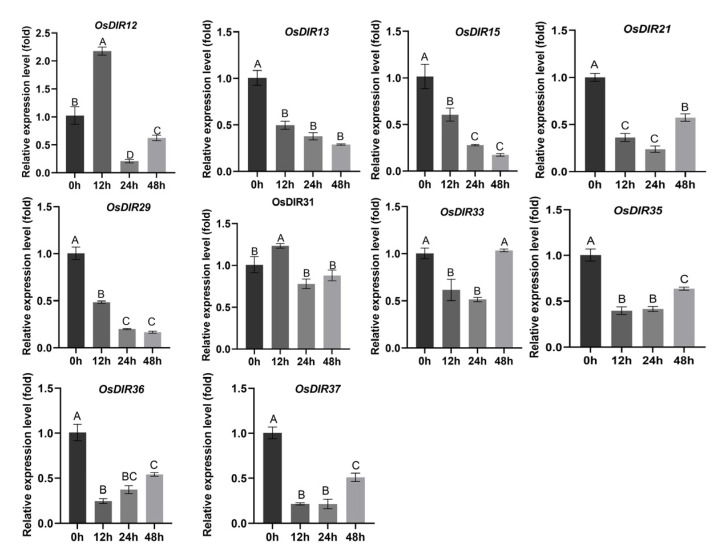
Expression patterns of 10 selected *OsjDIR* genes in response to *R. solani* inoculation treatment. All dates are the means ± SD of three biological replicates. Different letters indicate significant differences found via one-way ANOVA analysis (*p* < 0.01). qRT-PCR analysis was performed using the rice *OsActin* as an internal control.

**Figure 8 ijms-24-07189-f008:**
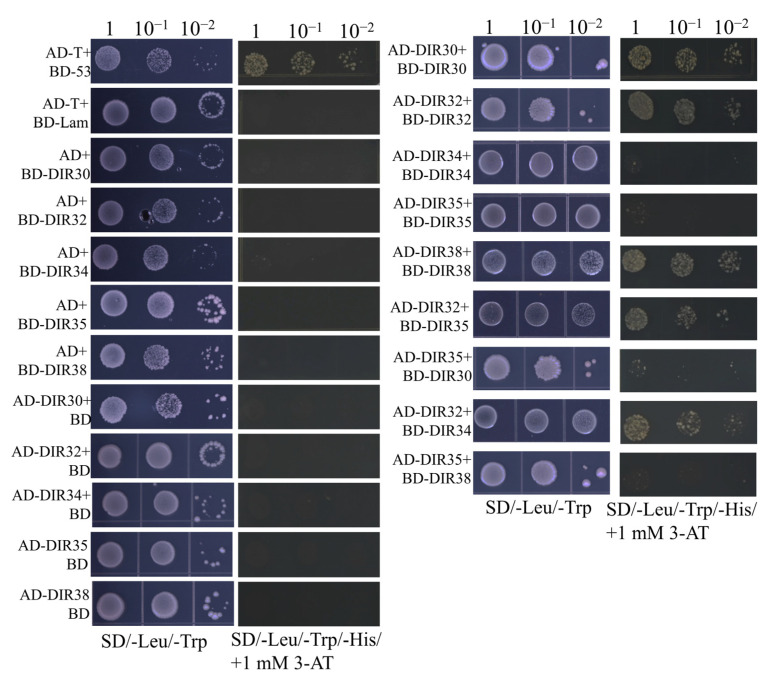
Yeast two-hybrid assays were used to identify the interactions between OsjDIR proteins. The positive control: AD-T + BD-53; the negative controls: AD-T + BD-Lam, AD+BD-DIRs and AD-DIRs+BD. Yeast transformations of the AD-DIRs prey and the BD-DIRs bait were performed via growth on SD/-Leu/-Trp, and the interaction was detected on SD/-Leu/-Trp/-His containing 1 mM 3-AT. Yeast transformations diluted in a dilution series (1, 10^−1^ and 10^−2^) showed in all pictures were plated on SD/-Leu/-Trp or SD/-Leu/-Trp/-His with 1 mM 3-amino-1,2,4-triazole (3-AT) after growth for 3 d at 28 °C.

**Table 1 ijms-24-07189-t001:** Summary of *DIR* genes in the genus *Oryza*.

Species	TD	SD	DIR-a	DIR-c	DIR-e	DIR-f	DIR-g	DIR-b/d	Number
*Oryza sativa* ssp. *indica*	18	6	7	12	6	7	8	8	48
*Oryza sativa* ssp. *japonica*	17	8	7	15	7	6	10	11	55
*Oryza glaberrima*	12	6	4	9	6	4	10	3	36
*Oryza nivara*	16	3	6	8	4	6	6	9	39
*Oryza rufipogon*	13	3	7	8	5	6	8	7	40
*Oryza barthii*	18	4	6	8	5	5	6	11	41
*Oryza glumipatula*	11	6	5	8	7	6	7	11	44
*Oryza meridionalis*	14	2	5	13	4	5	7	7	41
*Oryza punctata*	26	4	9	4	6	5	8	11	43
*Oryza brachyantha*	15	0	4	5	4	3	7	10	33

Tandem duplication (TD); segmental duplication (SD); *Oryza sativa* including *Indica* and *Japonica.*

## Data Availability

Not applicable.

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
