# Peer review of "Genome-Wide Identification and Expression Pattern Analysis of Dirigent Members in the Genus Oryza"

_ijms, 2023, doi:10.3390/ijms24087189_

Round 1

Reviewer 1 Report

This manuscript described collection and phylogenic classification of the Dirigent (DIR) member genes in Genus Oryza. In this study, the authors collected a total of 420 putative DIR genes. Based on integrating analysis with genome and transcriptome data and molecular biology experiments, the authors clearly demonstrated that many DIR genes are associated with responses to nutrition uptakes including mineral elements and disease infection such as Rhizoctonia solani. As the authors mentioned, the results in this study could provide a good research foundation for further exploration of the DIR genes in Genus Oryza including cultivated rice. the authors carried out a lot of bioinformatics analysis (dry data) and molecular biological experiments (wet data) to obtain conclusion in this study. However, the authors should make several revisions in the current version of this manuscript described below.

1) The authors should describe how many DIR genes have been reported in other plant species including Arabidopsis and other crop species at the Introduction or Discussion sections in the text. These descriptions would be helpful understanding research backgrounds and significances in this study.

2) The authors should describe how to detect and collect DIR genes in ten Oryza species at the first part of the Result section. What thresholds or criteria did you use to collect the DIR genes for databases?  And, the authors should indicate cultivar names or accession numbers for each Oryza species in Table 1 or the Materials and Methods section. For an example, genome sequence and trascriptome data for Orysa sativa ssp. Japonica are derived from a rice cultivar Nipponbare?

3) You could revise the order of several experiment methods at the Materials and Methods section. Because the authors firstly indicated the results for identification and phylogenic analysis of DIR genes in this manuscript, you could indicate each detailed method according to the order of 4.3, 4.4, 4.5, 4.6, 4.1, 4.2, 4.7 and 4.8 etc.

Reviewer 2 Report

Dear authors!

Thank you for your great work.

The article is filled with modern research methods, but it contains a number of shortcomings and questions:

1) The 1st paragraph of the “Introduction” contains sentences that are inconsistent and unrelated to each other in terms of meaning. When reading it, it is impossible to understand why you are writing here about suberin, ligan, lignin, pinoresinol and lignans. This paragraph needs to be rewritten to make it clear to the reader.

2) Your phrase “Thus, the genes that contribute to yield, stress tolerance and infection resistance might be lost during rice domestication” is incorrect in my opinion. So you are saying that artificial selection took place with the preservation of plants with non-viable traits? This is contrary to the generally accepted theory of artificial selection.

4) Why did you choose Rhizoctonia solani as a phytopathogen, and not other phytopathogenic bacteria that cause rice diseases, from which yield losses are also high, for example, fungi Fusarium spp., bacteria X. campestris pv. Oryzae, Erwinia chrysanthemi, Burkholderia glumae, Pseudomonas fuscovaginae?

4) In the "Results" section, you write: "The total numbers of DIR 92 proteins in wild rice species are significantly less than those in cultivated rice Oryza sativa." How do you explain this fact? Explain in the text of the article.

5) Please explain how the transcription of the DIR genes can respond to infection with a pathogen. What defense mechanisms are activated and proteins are synthesized? Explain in the text of the article.

6) You write: "Also note that the transcript abundance of OsjDIR12 gene was significantly up-regulated at 12 h after the inoculation of R. solani, whereas it was significantly down-regulated at 24 and 48 h post inoculation". What do you associate it with? Explain in the text of the article.

7) In the "Discussion" section, links to figures and tables are usually not written.

8) The description of the essence, results and significance of experiments using yeast is absolutely incomprehensible.

9) The "Materials and Methods" section does not indicate where the rice lines used in the studies were obtained.

10) For low nitrogen treatment of plants The roots were sampled at 2 and 7 day post treatment. Why did you choose exactly these observation points: 2 and 7 days?

11) The conclusion is too short. It needs to be expanded.

12) Of the 57 sources of literature, only 8 for the last 3 years. Please update the bibliography.

I recommend that the authors rewrite the Introduction, Discussion and Conclusion. Make them more understandable to the reader.

There is no information in Materials and Methods about statistical processing.

Respectfully Yours, reviewer.

April 04, 2023
